# Results of Pregnancy Control before and during the COVID-19 Pandemic: A Comparison of Two Cohorts

**DOI:** 10.3390/ijerph18158182

**Published:** 2021-08-02

**Authors:** Javier Llorca, Carolina Lechosa-Muñiz, Pelayo Frank de Zulueta, Sonia López-Gómez, Victoria Orallo, Jéssica Alonso-Molero, Barbara Arozamena-Llano, Yolanda Jubete, María Paz-Zulueta, María J. Cabero

**Affiliations:** 1Faculty of Medicine, Universidad de Cantabria, 39011 Santander, Spain; carolina.lechosa@scsalud.es (C.L.-M.); alonsomoleroj@gmail.com (J.A.-M.); maria.paz@unican.es (M.P.-Z.); mariajesuscabero@gmail.com (M.J.C.); 2CIBER Epidemiología y Salud Pública (CIBERESP), 28029 Madrid, Spain; 3Service of Paediatrics, Hospital Universitario Marqués de Valdecilla, 39008 Santander, Spain; pelayo.frank@scsalud.es (P.F.d.Z.); sonia.lopezg@scsalud.es (S.L.-G.); victoria.orallo@scsalud.es (V.O.); barbara.arozamena@scsalud.es (B.A.-L.); myolanda.jubete@scsalud.es (Y.J.); 4IDIVAL, Instituto de Investigación Sanitaria Valdecilla, 39011 Santander, Spain

**Keywords:** COVID-19, pregnant, cohort

## Abstract

The COVID-19 pandemic placed pregnant women at high risk, but behavioural changes have also led to lower rates of preterm births in high-income countries. The main goal of this article is to study the ongoing impact of the COVID-19 pandemic on pregnancy control and outcomes; this is a joint analysis of two cohorts. The pre-pandemic cohort includes 969 pregnant women recruited in 2018. The pandemic cohort comprises 1168 pregnant women recruited in 2020. Information on demographic and socioeconomic characteristics, reproductive history, characteristics of the current pregnancy and its outcome were obtained from medical records. Birth by Caesarean section was more frequent in the pre-pandemic cohort (adjusted odds ratio (OR) = 0.71, 95% confidence interval (CI): 0.55–0.92). A birth weight lower than 2500 g and higher than 4000 g occurred more frequently in the pre-pandemic cohort (adjusted OR = 0.62, 95% CI: 0.41–0.93 for lower than 2500 g and adjusted OR = 0.30, 95% CI: 0.20–0.46 for higher than 4000 g). Exclusive breastfeeding upon hospital discharge was more frequent in the pandemic cohort than in the pre-pandemic cohort (60% vs. 54%, *p* = 0.005), with adjusted OR = 0.67, 95% CI: 0.52–0.86 for mixed breastfeeding and infant formula. In conclusion, we report reductions in Caesarean sections and reduced numbers of low birth weight babies during the pandemic in a hospital located in northern Spain. Further analysis will clarify if these reductions are related to changes in health-related behaviour or healthcare operation.

## 1. Introduction

Infection by SARS-CoV-2 places pregnant women at high risk [1], as previously found in other highly pathogenic coronaviruses, such as SARS and MERS. The COVID-19 pandemic, however, can affect pregnancy not only via maternal infection, but also by its impact on social functioning, for e.g., national lockdowns and stay-at-home orders; health-related behaviour, such as working from home; and health care disruption, including telemedicine, curtailed provision of obstetric services, and weakened healthcare-seeking behaviour.

While some effects of the COVID-19 pandemic on pregnancy are deleterious, for e.g., higher maternal morbidity and mortality, preterm births have decreased in high-income countries, although a meta-analysis found important heterogeneity among studies [2].

Spain has been one of Europe’s most severely affected countries by COVID-19. A partial lockdown was declared on 14 March 2020 and complete lockdown, which halted all non-essential activity, was declared from 29 March to 12 April 2020. After 21 June 2020, regional rather than national restrictions were applied [3].

To further study the ongoing impact of the COVID-19 pandemic on pregnancy control and outcomes, we compared two cohorts of pregnant women recruited in 2018 and 2020 at the University Hospital Marqués de Valdecilla (HUMV), in Santander, northern Spain.

## 2. Materials and Methods

### 2.1. Setting and Patients

This study is a joint analysis of two cohorts recruited at the HUMV, Santander, Spain. A pre-pandemic cohort was recruited from 1 January to 31 August 2018, which included 969 pregnant women and their children. The main characteristics are described elsewhere [4].

The pandemic cohort was recruited in 2020. Its profile has already been reported [3]. Recruitment began on 26 May 2020 and finished on 22 October 2020. This cohort comprised three sub-cohorts. Sub-cohort 1 was retrospectively recruited and includes women delivering at HUMV between 23 March 2020 (the first day that the hospital introduced routine SARS-CoV-2 infection tests via PCR for all women admitted for delivery) and 25 May 2020. Sub-cohort 2 was prospectively recruited and includes women delivering at HUMV from 26 May and onwards. Sub-cohort 3 was prospectively recruited and included women who regularly attended HUMV for their routine 12-week pregnancy consultation. Many of their pregnancies are still ongoing at the time of this analysis. The rationale for these three sub-cohorts was to differentiate the pandemic’s consequences on pregnancy according to the risk period for each woman, which are classified as follows: Women in the first sub-cohort were neither exposed to or aware of SARS-CoV-2 for most of their pregnancy, and their exposure was limited to the last trimester, mostly coinciding with the first pandemic wave. Women in the second sub-cohort may have been exposed to and aware of the pandemic from their second trimester of pregnancy, which coincided with the first pandemic wave, while their third trimester was concurrent with lower levels of COVID-19 incidence between the first and the second waves. Finally, women in the third sub-cohort were aware of the pandemic throughout their pregnancy, and their exposure was higher in the second and subsequent waves.

### 2.2. Gathered Information

Information on maternal age, educational level (classified as primary school, secondary school, vocational training and university), occupation status (classified as employed, unemployed or inactive, student), number of previous children, length of pregnancy (later classified as less than 34 weeks, 34–36 weeks + 6 days, 37 weeks or more), type of delivery (eutocic, instrumental or Caesarean section), weight at birth (later classified as less than 2500 g, 2500–4000 g and more than 4000 g), and type of feeding provided upon hospital discharge (exclusive breastfeeding, mixed breastfeeding and infant formula, and infant formula alone) were obtained from medical records.

### 2.3. Statistical Analysis

In this article, we compare the pre-pandemic and pandemic cohorts. The description for each variable is provided as a number with a percentage or mean with standard deviation. The association between ordinal variables (i.e., age at delivery and educational level) and cohorts was tested using the Goodman–Kruskal γ test; its results are reported as γ, asymptotic standard error and *p* value. The association between categorical variables (i.e., occupation status and birth order) and cohorts was tested using χ^2^ analysis of contingency tables; its results are provided as χ^2^ statistics, number of degrees of freedom and *p* value. The association between pregnancy results (i.e., type of delivery, length of pregnancy, weight at birth and feeding upon hospital discharge) and cohorts was analysed using logistic regression, adjusting for age at delivery, educational level, and occupationalstatus. Models on the type of delivery were also adjusted for the length of the pregnancy. When the outcome was trichotomic, two models were obtained, omitting a category of the outcome each time. For instance, when studying a type of delivery, one model was obtained by comparing Caesarean section with eutocic delivery, and another model compared instrumental with eutocic delivery. Logistic regression results are displayed as odds ratios (OR) with 95% confidence intervals and *p* values. The pre-pandemic cohort was used as reference; therefore, OR > 1 indicates that the studied outcome was more frequent in the pandemic cohort, and OR < 1 indicates the opposite. All *p* values are two-tailed. The statistical analysis was conducted with the package Stata 16/SE (StataCorp, College Station, TX, USA).

## 3. Results

In this study, 2137 pregnant women were included, with 969 belonging to the pre-pandemic cohort and 1168 women to the pandemic cohort. In the pandemic cohort, 270 women who delivered a baby before 26 May 2020 were retrospectively recruited (sub-cohort 1); 350 women were prospectively recruited at delivery from 26 May 2020 (sub-cohort 2), and 548 women were prospectively recruited at week 12 of pregnancy (sub-cohort 3). Only 53 women in sub-cohort 3 already delivered a baby by the time of this analysis. Therefore, this sub-cohort was excluded from the variables related to delivery.

Table 1 provides a description of the participants. Approximately 80% of women in this study are aged 28–40 years, without differences between pre-pandemic and pandemic cohorts (*p* = 0.94). Women in the pre-pandemic cohort had lower educational attainment, 34% having received secondary schooling or lower vs. 21% in the pandemic cohort (*p* < 0.001). Pregnant women in the pre-pandemic cohort were more frequently unemployed or inactive (30%) than in the pandemic cohort (24%) (*p* = 0.009). There were no differences between the cohorts regarding the current pregnancy’s birth order (*p* = 0.42).

Table 2 displays results regarding delivery. Both instrumental and Caesarean section were less frequent in the pandemic cohort (adjusted OR = 0.66, 95% CI: 0.45–0.99 and adjusted OR = 0.71, 95% CI: 0.55–0.92, respectively). Prematurity, defined as a pregnancy length under 37 weeks, was slightly less frequent in the pandemic cohort, although far from being significant (adjusted OR = 0.78, 95% CI: 0.50–1.21). Low weight at birth, defined as less than 2500 g, was less frequent in the pandemic cohort (adjusted OR = 0.62, 95% CI: 0.41–0.93). Birth weight higher than 4000 g was more frequent in the pre-pandemic cohort (adjusted OR = 0.30, 95% CI: 0.20–0.46). Finally, mixed breastfeeding and infant formula upon hospital discharge was less frequent in the pandemic cohort (adjusted OR = 0.67, 95% CI: 0.52–0.86), while we could not find differences in the use of infant formula alone.

## 4. Discussion

According to our results, Caesarean section, and instrumental delivery were less frequent in the pandemic year than two years before. Low weight at birth was also less frequent in the pandemic cohort. These results are noteworthy considering that the previously reported trend of low birth weight in Spain in the 21st century has been towards a fast increase [5]. Explanations for these results may be related to changes in healthcare during the pandemic, greater self-protective behaviour by pregnant women, and stay-at-home orders.

Previous studies have reported contradictory results regarding preterm birth or low weight at birth in developed countries. Dramatic decreases in extremely premature (i.e., gestational age at birth under 28 weeks) have been reported in a Danish population [6], as well as in Ireland regarding very low birth weights and extremely low birth weights [7]. In Italy, however, only a slight decrease was found in late preterm births [8]. Handley et al. (2021) [9], on the other hand, did not find changes in preterm rates in Philadelphia. Main et al. (2020) [10] described a slight increase in preterm births between 28 and 31 + 6 weeks in California, but no changes in other gestational ages. The greater part of this increase appeared in Hispanic/Latino populations. It is noteworthy that European studies tend to describe the decline in preterm rates, whereas American studies describe no change or small increments. Pandemic-associated factors that may explain differences between studies can include changes in healthcare and a pregnant woman’s self-protective behaviour.

Changes to healthcare operation during the pandemic include reductions in antenatal maternity consultations, a rise in remote appointments and a diminution in emergency antenatal presentations [11]. Healthcare services in western Europe are usually public instead of private and have universal coverage. Further information is required to ascertain if attending to pregnancy has changed in European countries from the US during the pandemic, including access to health care.

Health-related behaviour in pregnant women may have changed more protectively, as a response to the perceived risk COVID-19 would place on them. For instance, less social activity and lower physical demands, including, among others, earlier maternity leave and more time to expend at home, may have led to lower foetal stress. On this subject, studies on the putative connection between work and prematurity are contradictory [12]. National lockdowns and stay-at-home orders may have also contributed to changes in women’s health-related behaviour. However, the contribution of lockdowns may vary according to the restriction levels and how they were reinforced by authorities [10].

Reported changes in Caesarean section rates appear to be a small amount and with unpredictable direction. Thus, a small increase was reported in England (from 28.3 to 29.7%) [13], a nonsignificant decrease in Italy (from 36.2 to 35.5%) [8], and no relevant changes in New York (from 31.7 to 31.3%) [14]. In our results, Caesarean section rates were largely cut down from 24 to 18% in just two years. This decrease cannot be wholly attributed to changes during the pandemic period. In reality, two factors played a role in this decrease: Firstly, to diminish Caesarean rates was an institutional target before the pandemic began. Secondly, women infected with COVID-19 are considered a high surgical risk; therefore, they have been closely monitored, and induction was used early on to avoid Caesarean section. We have no data to evaluate the relative contribution of these two factors to the decreasing rate of Caesarean sections. However, it is noteworthy that a relevant part of it may be related to changes in women’s pregnancy characteristics and pregnancy length in the pre-pandemic to the pandemic cohort, as shown by the attenuation of odds ratio when those variables were adjusted.

Apart from changes in health-related behaviour and the health care system’s performance, differences in pregnancy outcomes between the pre-pandemic and pandemic cohorts may have been a direct effect of infection by SARS-CoV-2. In our pandemic cohort, however, only 37 women tested positive with SARS-CoV-2 infection whether via PCR test or antibody detection [15]. Positivity was not associated with educational level, occupation status or pregnancy outcome [3,15]. Changes in pregnancy outcome as a direct result of SARS-CoV-2 infection (if they occur) are expected to be deleterious. However, most of the changes to pregnancy outcome that we report were towards improving results in the pandemic cohort (i.e., less instrumental deliveries and Caesarean rates, fewer low birth weights). Therefore, it is hardly credible there is a causal relationship with coronavirus infection during pregnancy.

Our study has some limitations. Firstly, it was conducted in a single hospital. This is a double-edged characteristic: On the one hand, it is difficult for us to generalize our results; on the other hand, it allows us to collect data in a reliable, standardized way. Secondly, most of the data we are reporting on the pandemic cohort includes women who were already pregnant when the pandemic was declared. Sub-cohort 3, comprising women who became pregnant after the pandemic began, can help clarify some remaining questions, including the relevance of healthcare changes throughout their whole pregnancy. Thirdly, the sample size of our cohorts is limited in order to ascertain rare pregnancy outcomes, such as extreme prematurity. Finally, we did not collect data on prenatal counselling in the pre-pandemic cohort. Therefore, we cannot rule out that counselling may be responsible for some of the pregnancy outcomes. However, counselling in the pandemic cohort occurred only in 228 women (19%). We could not have expected this low figure to be higher in the pre-pandemic cohort, when the healthcare system was performing normally, as prenatal counselling is usually offered to all pregnant women in our healthcare system.

## 5. Conclusions

In conclusion, we report a reduction in Caesarean sections and preterm birth during the pandemic in a hospital located in northern Spain. Further analysis would clarify if these declining rates are connected to changes in health-related behaviour or healthcare operation.

## Figures and Tables

**Table 1 ijerph-18-08182-t001:** Description of the cohorts included in this analysis.

Variable	Pre-Pandemic Cohort (Recruited in 2018); n = 969	Pandemic Cohort (Recruited in 2020) n = 1168	Goodman–Kruskal γ Test (Asymptotic Standard Error) or χ^2^ Test (Degrees of Freedom)	*p* Value Between Pre-Pandemic and Pandemic Cohorts
**Age at delivery**				
<24 years	36 (4)	38 (3)	γ = 0.0027 (0.035)	0.94
24–27 years	85 (9)	113 (10)		
28–34 years	411 (42)	476 (41)		
35–40 years	363 (37)	468 (40)		
>40 years	74 (8)	73 (6)		
**Educational level**				
Primary school	215 (22)	175 (15)	γ = 0.2023 (0.033)	<0.001
Secondary school	114 (12)	73 (6)		
Vocational training	281 (29)	369 (32)		
University	359 (37)	544 (47)		
**Occupational status**				
Employed	673 (69)	865 (75)	χ^2^ = 9.34 (2)	0.009
Unemployed/Inactive	286 (30)	277 (24)		
Student	10 (1)	18 (2)		
**Birth order**				
First	507 (52)	335 (54)	χ^2^ = 0.64 (1)	0.42
Other	462 (48)	281 (46)		

**Table 2 ijerph-18-08182-t002:** Type of delivery, length of pregnancy, and weight at birth in the two cohorts. In the pandemic cohort, only sub-cohorts 1 and 2 are included, as most pregnancies in sub-cohort 3 are still ongoing.

Variable	Pre-Pandemic Cohort (Recruited in 2018); n = 969	Pandemic Cohort (Recruited in 2020); n = 620	uOR (95% CI)	*p* Value	aOR (95% CI)	*p* Value
**Type of delivery**						
Eutocic	653 (67)	455 (75)	1 (ref.)	-	1 (ref.)	-
Instrumental	80 (8)	40 (7)	0.75 (0.51–1.10)	0.14	0.66 (0.45–0.99)	0.05
Caesarean section	236 (24)	110 (18)	0.34 (0.27–0.43)	<0.001	0.71 (0.55–0.92)	0.009
**Length of pregnancy**						
<34 weeks	20 (2)	3 (1)				
34–366 weeks	39 (4)	23 (4)	0.85 (0.56–1.31) *	0.47 *	0.78 (0.50–1.21) *	0.26 *
≥37 weeks	910 (94)	573 (96)	1 (ref.)	-	1 (ref.)	-
**Weight at birth**						
<2500 g	83 (9)	34 (6)	0.62 (0.42–0.92)	0.02	0.62 (0.41–0.93)	0.02
2500–4000 g	808 (83)	550 (90)	1 (ref.)	-	1 (ref.)	-
>4000 g	78 (8)	25 (4)	0.28 (0.19–0.43)	<0.001	0.30 (0.20–0.46)	<0.001
**Feeding at hospital discharge**						
Exclusive breastfeeding	521 (54)	359 (60)	1 (ref.)	-	1 (ref.)	-
Mixed breastfeeding and infant formula	280 (29)	128 (22)	0.65 (0.51–0.83)	<0.001	0.67 (0.52–0.86)	0.002
Infant formula	168 (17)	107 (18)	0.96 (0.73–1.23)	0.68	1.03 (0.78–1.37)	0.83

uOR: unadjusted odds ratio; CI: confidence interval; aOR: odds ratio adjusted for age at delivery, educational level, and occupation status (models on length of pregnancy, weight at birth, and feeding at hospital discharge). Type of delivery model was adjusted for length of pregnancy. * Combining < 34 and 34–36^6^ categories.

## Data Availability

The data presented in this study are available on request from the corresponding author. The data are not publicly available due to patients’ privacy.

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
