# Peer review of "Results of Pregnancy Control before and during the COVID-19 Pandemic: A Comparison of Two Cohorts"

_ijerph, 2021, doi:10.3390/ijerph18158182_

Round 1
Reviewer 1 Report
The study seems simple, but the effort of field work, recruitment and permanent registration of information is valued.
The manuscript is well writen.
It would seem that presenting general information from the third subcohort is too much, since the birth results are not presented.
It remains to discuss the differences between the groups in terms of education and occupation, to propose some explanations or hypotheses about this observation.
It would be of interest to know the data on attendance at prenatal check-ups in both cohorts, since they also mention it in the discussion, but they are not presented.
Author Response
Attached is an item-by-item response to all comments raised by the editor and the reviewers.

Reviewer 2 Report
I have read the paper of Lorca and colleagues. While I find the paper interesting for the topic addressed I find the way of the presentation of study design and results as being poor.
Unfortunately, as the same authors show in Table 1, I find out the two samples of being very different: educational status/occupational status.
Thus the results may be biased either by these reasons, and also by other not explored areas like the attendance in pregnancy, the use of counseling centers, etc.
In addition this paper is badly written with lots of typos (some example: "bay" intead of "baby", covid-19 intstead of COVID-19) and bad english language.
Author Response

(The authors gave the same response as above.)

Reviewer 3 Report
The study is interesting and because of the COVID-19 pandemic, its results are highly relevant to the scientific and health community. However, all the investigations are based on the results, and in that sense, I consider that the following points must be addressed for the manuscript to be published.
- The design corresponds to a factor (Pandemic cohor) with three levels and characteristics. In this sense, the authors indicated that the counts were analyzed with the Chi-square test. However, it is a very common error seen in many articles where they only mention the test under the wrong name. Since they must refer and check if it corresponds to contingency tables with a Chi-square distribution for counting variables (ordinal), which seems to indicate that it does. For example, 3x5 Contingency Table (Chi-Square Analysis of Contingency Table) where the columns correspond to the columns: delivery before 26th May, recruited at delivery from 26th May, and recruited at 12th week consultation from 26th May by five age levels. Therefore, I suggest you review your evidence and indicate it appropriately. Furthermore, the Chi-square values and degrees of freedom are not indicated in the results, and there are only the P values. Therefore, it is pertinent to indicate the three parameters of the tests. Below the column that indicates the value of P in Tables 1 and 2, there is enough space to place the values.
- It is indicated that they performed Student's t test lines 93-94. However, as they did not locate the value of t and its degrees of freedom, the reading is confused as to which result it corresponds. Also, if the authors performed this test, they should also indicate whether they performed the verification of the homogeneity of variances and normal error distribution of the response variable. If they are not fulfilled, they must indicate the procedure to correct this effect. It is important to indicate that these assumptions were verified, otherwise, it increases the probability of committing type II statistical error.
I must indicate that the study is based on the results presented, therefore, avoiding confusion and a good application of statistical methods is essential for the reproducibility of the study. In addition, it allows greater robustness for the projection of the inferences indicated in the discussion.
Author Response

(The authors gave the same response as above.)

Round 2
Reviewer 2 Report
The paper is now improved particularly from a stylistica form. I would like to thank the researchers for this work.
Author Response
Thank you.
Reviewer 3 Report
Based on the reading of the manuscript in its new corrected and adjusted version. It is appropriate that the new statistical analyzes were carried out and they were indicated appropriately in the method and in the results Table 1 and 2. In addition to eliminating information that did not correspond to as Student's t-test. The language setting is very adequate eliminating errors. I consider that the manuscript in its current status is feasible for publication.
Author Response
Thank you.